# The Mechanism of DNA Methylation and miRNA in Breast Cancer

**DOI:** 10.3390/ijms24119360

**Published:** 2023-05-27

**Authors:** Lingyuan Ma, Chenyu Li, Hanlin Yin, Jiashu Huang, Shenghao Yu, Jin Zhao, Yongxu Tang, Min Yu, Jie Lin, Lei Ding, Qinghua Cui

**Affiliations:** 1Lab of Biochemistry & Molecular Biology, School of Life Sciences, Yunnan University, Kunming 650091, China; malingyuan15@163.com (L.M.); 15712763321@163.com (C.L.); 15823076461@163.com (H.Y.); huangjiashu_666@163.com (J.H.); ysh1042939168@163.com (S.Y.); zj651353@163.com (J.Z.); yxtang899@126.com (Y.T.); yumin@ynu.edu.cn (M.Y.); linjie@ynu.edu.cn (J.L.); 2Yunnan Collaborative Innovation Center for Plateau Lake Ecology and Environmental Health, Kunming 650214, China

**Keywords:** breast cancer, DNA methylation, miRNA, drug resistance

## Abstract

Breast cancer is the most prevalent cancer in the world. Currently, the main treatments for breast cancer are radiotherapy, chemotherapy, targeted therapy and surgery. The treatment measures for breast cancer depend on the molecular subtype. Thus, the exploration of the underlying molecular mechanisms and therapeutic targets for breast cancer remains a hotspot in research. In breast cancer, a high level of expression of DNMTs is highly correlated with poor prognosis, that is, the abnormal methylation of tumor suppressor genes usually promotes tumorigenesis and progression. MiRNAs, as non-coding RNAs, have been identified to play key roles in breast cancer. The aberrant methylation of miRNAs could lead to drug resistance during the aforementioned treatment. Therefore, the regulation of miRNA methylation might serve as a therapeutic target in breast cancer. In this paper, we reviewed studies on the regulatory mechanisms of miRNA and DNA methylation in breast cancer from the last decade, focusing on the promoter region of tumor suppressor miRNAs methylated by DNMTs and the highly expressed oncogenic miRNAs inhibited by DNMTs or activating TETs.

## 1. Introduction

Breast cancer (BC) is the most frequently diagnosed cancer and the leading cause of cancer death in women worldwide [1]. BC can be divided into five subtypes based on the expression of estrogen receptor (ER), progesterone receptor (PR), human epidermal growth factor receptor-2 (HER-2) and other biomarkers—Luminal A, Luminal B, HER2-enriched, basal-like and a Normal Breast-like Group [2,3]. Hormonal therapy is commonly used for ER-positive breast cancer patients, but it usually causes the development of drug resistance [4], and the newly adjuvant chemotherapy can reduce mortality in breast cancer patients [5]. Targeted therapy is the most effective regimen recognized for the treatment of HER2+ breast cancer [6]. Case in point, trastuzumab, the first FDA-approved recombinant antibody, is used extensively to target HER2-positive breast cancer [7]. Immunotherapy has been proven to ameliorate patient prognosis and survival as well [8]. Nevertheless, the combination of chemotherapy and radiotherapy is the most important treatment for triple negative breast cancer (TNBC) defined as ER-, PR- and HER2-negative [8,9]. The therapy combination faces challenges due to the high heterogeneity, high rates of metastasis, poor prognosis and lack of therapeutic targets for TNBC [10]. Therefore, the pathogenesis of BC needs more exploration. Currently, the role of epigenetic alterations, such as DNA methylation and microRNA (miRNA), has become a fascinating area in relation to regulating gene expression at the pre- or post-transcriptional levels, which might offer a new opportunity for cancer clinical management in BC. In this review, we aim to summarize the roles of DNA methylation and miRNA in BC initiation, development and clinical therapeutics, and to review the relationship between miRNA and DNA methylation to identify potential therapeutic targets for breast cancer.

## 2. The Dysregulation of DNA Methylation in Breast Cancer

Epigenetics is a heritable molecular mechanism that regulates gene expression without altering the actual sequence of DNA, and it includes epigenetic regulations caused by DNA methylation, histone modification, chromatin remodeling and RNA-mediated gene targeting [11]. The alterations of epigenetics can regulate a number of molecular, cellular and biological pathways associated with breast carcinogenesis [12]. Studies have shown that the causes of thyroid cancer [13], lung cancer [14], liver cancer [15], gastric cancer [16], prostate cancer [17], bladder cancer [18], ovarian cancer [19], colorectal cancers [20] and breast cancer [21] are associated with abnormal methylation, and it can be said that almost all cancers are associated with abnormal DNA methylation. In this article, we focus on the abnormal regulation of DNA methylation in BC. DNA methylation involves adding methyl groups to cytosine via three DNA methyltransferases (DNMTs)—*DNMT1*, *DNMT3A* and *DNMT3B*—without changing the DNA sequence [22,23]. Abnormal methylation includes local hypermethylation of the promoter region in the CpG island of a specific gene and whole-genome hypomethylation in the genomic repeat regions [24] associated with malignant tumors [25,26], or the specific hypomethylation of some genes [24], such as signal-induced proliferation-associated protein 1 (*SIPA1*) [27].

### 2.1. DNA Hypermethylation in Breast Cancer

In mammals, methylation sites are primarily located on the CpG island in the promoter regions of genes [28]. When methylation occurs in the promoter region of a gene, transcription is inhibited directly by blocking the binding sites for the transcription factor or by recruiting methyl–CpG binding proteins [29]. Large amounts of evidence have shown that the DNA methylation level in BC cells is dramatically increased compared to that in normal cells. The hypermethylation of tumor suppressor genes (TSGs) promotes uncontrolled cell proliferation, resulting in metastasis, which can be used as a biomarker for breast cancer diagnosis and treatment [30,31,32]. Glutathione S-transferase Mu 2 (*GSTM2*) has been identified in aggressive, high-grade breast tumors where promoter hypermethylation is associated with ER/PR-negative status among ductal carcinoma in situ (*DCIS*) and invasive tissue components [33]. The expression of the tumor suppressor gene superoxide dismutase3 (*SOD3*) is downregulated in breast cancer; therefore, *SOD3* expression levels are inversely correlated to its promoter CpG methylation, which is significantly associated with poor outcome patients [34].

### 2.2. DNA Hypomethylation in Breast Cancer

Regional DNA hypomethylation also occurs in cancer, and the frequency of this is lower than that of regional DNA hypermethylation [35]. A reduced probability of cytosine being methylated during the global DNA hypomethylation of tumor genomes corresponds to an average loss of about 5% to 20% of 5mC bases [36]. In the early stages of cancer, the DNA hypomethylation of non-coding repetitive elements is a shared trait of tumor cells [37]. DNA hypomethylation occurs in about 50% of breast cancers [38,39], and correlates to histologic grade stage and malignancy [40]. DNA methylation is regulated by the balance of DNMTs and DNA demethylases (TETs). Two patterns of hypomethylation occur in BC—one resulting in the hypomethylation of aberrant oncogenes by demethylases, and the other resulting in reduced levels of oncogene methylation due to the absence of DNA methylation. The TETs (*TET1*, *TET2* and *TET3*) are DNA demethylation enzymes that convert 5 methyl-cytosine (5mC) into 5 hydroxymethylcytosine (5hmC) [41].

In TNBCs, *TET1* causes DNA hypomethylation and the activation of oncogenic signaling pathways [42]. Estrogen (17β estradiol; E2) induces *DNMT3B*-mediated hypomethylation in the promoter of the Yes-associated protein-1 (*YAP1*) and augments proliferation [43]. The overexpression of the potassium two pore domain channel subfamily K member 9 (*KCNK9*/*TASK3*) protein by the hypomethylation of *KCNK9* differential methylation region (DMR) elevates mitochondrial membrane potential and inhibits apoptosis [44]. The increased expression of signal-induced proliferation-associated protein 1 (*SIPA1*) caused by the hypomethylation of the CpG island in the promoter-proximal element ultimately promotes epithelial–mesenchymal transformation (EMT) [27]. The methylation of the ADAM metallopeptidase domain 12 (*ADAM12*) in TNBC is lower than that in Non-Neoplastic Breast Tissues. Hypomethylation could be a poor outcome biomarker and a potential therapeutic target [45]. Long-disseminated non-coding element 1 (*LINE-1*) hypomethylation is reported as a biomarker for patients with low-grade BC [46].

## 3. Aberrant Methylation Associated with Drug Resistance in Breast Cancer

Chemotherapy, surgery and radiation therapy are currently the main pillars of cancer treatment [47]. However, drug resistance due to abnormal methylation limits the efficacy of these therapies [48].

### 3.1. Hypermethylation of Genes Associated with Drug Resistance in Breast Cancer

The changes in the DNA methylation status of the promoters of certain genes correlate with drug resistance in breast cancer (as shown in Table 1). The hypermethylation of glucosylceramide synthase (GCS) enhances drug resistance [49]. In patients with ER-positive breast cancer, endocrine therapy works by blocking the attachment of estrogen to ER. Endocrine resistance is related to the methylation of estrogen receptor 1 (*ESR1*) in cell-free DNA (cfDNA) [50]. The hypermethylation of the spalt-like transcription factor 2 (*SALL2*) promoter results in tamoxifen resistance [51]. The abnormal DNA methylation of bone morphogenetic protein 6 (*BMP6*), crucial to the EMT phenotype, contributes to doxorubicin resistance [52]. In HER2-positive breast cancer patients, the first targeted drug to be approved is trastuzumab [53]; however, a large number of patients have shown primary or acquired resistance, which is significantly related to the hypermethylation of the tumor suppressor gene transforming growth factor beta induced (*GFBI*) promoter [54]. Oftamoxifen resistance is associated with the methylation aberration of the paired box 2 (*PAX2*) promoter [55]. ERα methylation develops cisplatin resistance [56], while docetaxel resistance is caused by the methylation of the Ras association domain family member 10 (*RASSF10*) [57].

### 3.2. Hypomethylation of Genes Associated with Drug Resistance in Breast Cancer

Epigenetic changes in breast cancer, including altered levels of methylation at the CpG site, are normally located on the promoter of the genes, and have been linked with resistance to some chemotherapeutic agents (as shown in Table 2). In the MCF-7 cell line, tamoxifen resistance is caused by the hypomethylation of CpG islands of the lactate dehydrogenase B (*LDHB*) promoter [58]. In the tumor and serum of patients with invasive ductal breast cancer, increases in tumor size and advanced tumor stage were significantly correlated with the hypomethylation of Multi-Drug Resistance Gene-1 (*MDR1*) [59]. The hypomethylation of matrix metallopeptidase 1 (*MMP1*) may be the cause of tamoxifen resistance in breast cancer. In tamoxifen-resistant MCF-7 cells, it was found that *MMP1* was hypomethylated and over-expressed. The downregulation of *MMP1* enhances sensitivity to tamoxifen and increases cell apoptosis [60]. Studies have shown that, compared with the TNBC samples before treatment, the hypomethylation of the SH3 domain containing GRB2-like 2 (*SH3GL2*) promoter in neoadjuvant chemotherapy-treated (NACT) samples leads to the downregulation of epidermal growth factor receptor (EGFR) protein, and finally, the nuclear expression of Y654-p-β-catenin in NACT samples decreases, resulting in a low proliferation index/CD44 prevalence rate. Therefore, the hypomethylation of the *SH3GL2* promoter plays an important role in the chemical tolerance of TNBC [61].

## 4. MiRNA and DNA Methylation

MiRNAs are small non-coding RNA with a length of 18 to 25 nucleotides [62]. MiRNA biogenesis involves three main steps: firstly, the fragment located in an intragenic region or gene desert is transcribed by RNA polymerase II (*pol II*), which produces an mRNA with a length of about 3000 to 5000 bases, called primary miRNA (pri-miRNA). In the second step, pri-miRNA is shortened to produce a 70-base precursor miRNA (pre-miRNA), which is shortened by DROSHA and DGCR8. Finally, the export receptor export 5 (Exp5) directly interacts with the pre-miRNA to transport it into the cytoplasm, and then the Dicer cleaves the pre-miRNA into short fragments with a length of 22 bases. Then, its double strand is dissociated into a single strand. AGO-2 is connected with one of the mature strands to form an RNA-induced silencing complex (RISC) [62,63,64,65]. The process is as shown in Figure 1. According to the miRNAs’ progression- or suppression-promoting function, they can be used as oncogenes or as tumor suppressors, respectively [66]. One example is miR-125b, which is downregulated in breast-cancer-promoting cell proliferation and cell-cycle progression [67].

Under the action of pol II, the DNMTs gene undergoes intron shearing, exon splicing and a series of complex processes, before finally becoming mature messenger RNA (mRNA) [65], as shown in Figure 1. Studies have shown that *DNMT1*, *DNMT3A* and *DNMT3B* are usually highly expressed in patients with advanced breast cancer [63]. The hypermethylation of the CpG-rich promoter regions of tumor suppressor miRNAs usually leads to their silencing [68] through DNMTs. The hypomethylation of oncogenic miRNAs was downregulated in breast cancer. The abnormal DNA methylation of miRNA usually leads to the downregulation of miRNA, which is significantly related to the malignant phenotype of breast cancer cells [69]. In turn, miRNA could regulate the expression level of the target gene after transcription, which affects DNA methylation in breast cancer cells through the 3′ untranslated region (3′-UTR) of the RISC-targeted DNMTs’ mRNA [70]. In conclusion, this demonstrates the existence of a regulatory loop between miRNA expression and epigenetic modifications [71].

## 5. The Relationship between miRNA and DNA Methylation

In breast cancer, there are three patterns of miRNA and DNA methylation. The first is the epigenetic silencing of miRNAs to inhibit the transcription of miRNAs, by which DNMTs downregulate the expression of miRNAs through methylation in the promoter region of miRNAs, which are usually suppressor miRNAs. The second is that miRNAs inhibit the expression of DNMTs, where miRNAs (generally suppressor miRNAs) target the 3′-UTR region of DNMTs through RISC to inhibit the expression of DNMTs. The third type is the abnormal hypomethylation of oncogenic miRNAs that are highly expressed in breast cancer and thus promote its development.

### 5.1. Aberrant Methylation of Tumor Suppressor miRNA Promoter in Breast Cancer

In breast cancer, tumor suppressor miRNAs epigenetically silenced by DNMTs are usually involved in cell proliferation, migration, invasion and stemness, as well as cell apoptosis, and can be used as clinical markers (as shown in Table 3). MiR-29c inhibits tumor growth by regulating TGFB-induced factor homeobox 2 (*TGIF2*), CAMP-responsive element binding protein 5 (*CREB5*), and V-Akt murine thymoma viral oncogene homolog 3 (*AKT3*). The miR-29c is gradually downregulated via DNA methylation in the promoter region involved in the occurrence and development of tumors [72]. Flap structure-specific endonuclease 1 (*FEN1*) promotes miR-200a methylation by forming an FENl/PCNA/DNMT3A complex that inhibits the signal transduction of MET proto-oncogene, receptor tyrosine kinase (MET) and epidermal growth factor receptor (*EGFR*), thus repressing cell proliferation [73]. In aggressive breast cancer cell lines as well as untransformed mammary epithelial cells, the expression of the miR-200c/141 cluster is regulated by DNA methylation, and the epigenetic silencing of the miR-200c/141 cluster induces EMT [74]. The hypermethylation of the miR-203 promoter downregulates its expression in highly aggressive breast cancer cells. The overexpression of miR-203 inhibits tumor cell invasion by preventing the mesenchymal marker snail family transcriptional repressor 2 (*Snail2*) [75].

However, some other oncogenic miRNAs have multiple functions in breast cancer development. In a previous study, miR-195/497 was proven to have an inhibitory effect on breast cancer malignancies [76]. MiR-195/497 targets mucin1 (*MUC1*) and promotes the apoptosis of breast cancer cells by downregulating *MUC1*. In breast cancer tissues and cells, the methylation of CpG islands of the miR-195/497 promoter reduces the expression of miR-195/497 and promotes proliferation and invasion [77]. In TNBC, methylation on the CpG island of miR-296-5p and miR-512-5p improves the expression level of downstream target gene telomerase reverse transcriptase (*hTERT*), which is involved in inhibiting cell apoptosis and improving invasiveness [78]. MiR-31 inhibits metastasis and is highly expressed in early BC; as the tumor progresses to a more aggressive stage, the expression level of miR-31 is reduced and becomes undetectable in metastatic BC [79]. The loss of miR-31 expression in TNBC cell lines is attributed to promoter hypermethylation [80]. In breast cancer clinical samples and cell lines, promoter hypermethylation of miR-145 and direct targeting of the angiopoietin 2 (*AngpT2*) gene have been found. *AngpT2* is a member of the angiopoietin family that promotes tumor development and progression by linking metastatic inflammasomes to angiogenic processes. The methylation of miR-145 silences its expression and leads to the upregulation of *AngpT2*, promoting migration and invasion [81]. MiR-133a-3p is epigenetically suppressed and silenced by promoter methylation, which leads to a significant increase in the proliferation, migration, invasion and stemness of breast cancer cells in vitro, mainly through the miR-133a-3p/MAML1/DNMT3A positive feedback loop [82]. In the nucleus of the breast cancer cell, Kindlin 2 forms a complex with DNMT3A, which occupies the promoter of miR-200b to promote cell invasion migration and stemness [83]. The above-mentioned oncogenic miRNAs in breast cancer are usually down-regulated through epigenetic silencing, and have only oncogenic function according to the literature.

Epigenetically silenced miRNAs can not only affect the development of breast cancer, but also serve as clinical markers of breast cancer. The studies show that the high methylation of miR-124a-1/2/3 plays an important role in tumor growth and invasion [84]. MiR-132, miR-137 and miR-1258, with hypermethylation frequencies of 41%, 37% and 34%, respectively, were found to be associated with clinical features in a representative sample of 41 breast cancers [85]. Others have analyzed 91 representative samples of breast cancer and histologically normal tissues, showing that the hypermethylation of miR-9-3 and miR-339 is related to tumor size, and the hypermethylation of six miRNA genes (miR-124-1, miR-127, miR-34B/C, miR-9-3, miR-1258 and miR-339) was found to be significantly related to the late (III–IV) clinical stage [86].

There are also some epigenetically silenced miRNAs with unknown functions in breast cancer. The erb-b2 receptor tyrosine kinase 2 (*ErbB2*) signal induces the DNMTs family through the Ras/Raf/MEK/ERK pathway, leading to the hypermethylation of the CpG-rich region of the miR-205 promoter, which in turn inhibits the transcription of miR-205 [87]. In cervical cancer, miR-152 acts as an oncogene to promote the growth, survival, migration and aggressiveness of cancer cells [88]. In 71 primary human breast cancer specimens, the effect of demethylation on miRNA gene expression was that 34–86% of cases showed abnormal hypermethylation of miR-9-1, miR-124a3, miR-148, miR-152 and miR-663 [89]. MiR-9-1, miR-148, miR-152 and miR-663 can act as tumor suppressors in many cancers; miR-9-1 inhibits nasopharyngeal carcinoma growth and glucose metabolism [90]; miR-148 and miR-152 can inhibit the proliferation of prostate cancer cells [91]; and miR-663 suppresses the proliferation and invasion of colorectal cancer cells by directly targeting *FSCN1* [92].

### 5.2. The DNMTs Targeted by Tumor Suppressor miRNA to Deregulate DNA Methylation

In breast cancer, a high expression of DNMTs relates to the occurrence and development of breast cancer. Partially overexpressed tumor suppressor miRNAs could directly target and inhibit the expression of DNMTs, decreasing cell proliferation, migration, invasion and EMT (as shown in Table 4) and ultimately slowing down the development of breast cancer. MiR-152 targets *DNMT1* and downregulates the expression of *DNMT1*, which helps restore cadherin 1 (*CDH1*) gene expression and obstructs the migration of breast cancer cells [93]. MiR-148a regulates the DNA methylation of ER-α by targeting *DNMT1* [94]. Phosphatase and tensin homolog (*PTEN*), a tumor suppressor also methylated by *DNMT3A*, is a direct target of miR-143 [95]. In breast cancer tissues and breast cancer cell lines, the expression of miR-101 is downregulated and the expression of *DNMT3A* is upregulated. It has been found that miR-101 inhibits cell proliferation and migration by inhibiting *DNMT3A*, thus restoring the expression of E-cadherin [96]. MiR-194 targeted *DNMT3A* in BC, and the increased expression of miR-194 stimulated tumor suppressor cyclin G2, p27Kip1 and ADAM metalpeptidase domain 23 (*ADAM23*), leading to inhibited cell motility [97]. In TNBC, E-cadherin protein, encoded by *CDH1*, mediates intercellular adhesion. MiR-770-5p has been found to target and block the activity of *DNMT3A*, which is responsible for the hypermethylation of the *CDH1* promoter. The re-expression of miR-770-5p restores E-cadherin expression by targeting *DNMT3A*, thus reversing EMT to mesenchymal–epithelial transition (MET) [98]. In ER-positive breast cancer cells, miR-29c-5p downregulates *DNMT3A*, resulting in the hypomethylation of CpG in ER-related transcription factor binding site (*TFBS*). The CpG sites are located in *TFBS* enhancer regions of ER-α, forkhead box A1 (*FOXA1*) and GATA binding protein 3 (*GATA3*). All of these genes are key factors determining the phenotypes of luminal breast cancer [99]. In breast cancer cells, miR-29c targets and downregulates *DNMT3B* and reduces the DNA methylation level of TIMP metallopeptidase inhibitor 3 (*TIMP3*). Finally, the proliferation, migration, invasion, colony formation and growth of breast cancer are mediated by TIMP3/STAT1/FOXO1 [100]. MiR-148b, miR-29c and miR-26b target *DNMT3B* and prevent the expression of *DNMT3B* [101]. In human breast cancer stem cells (BCSC), miR-221 downregulates the expression of *DNMT3B* and changes the phenotype. *DNMT3B* inhibits the expression of Nanog and Oct 3/4 and increases the cell numbers. Therefore, the downregulation of *DNMT3B* may represent an advantage for cancer development and promote the expansion of stem cell compartment [102]. In non-invasive epithelial breast cancer cells, the low expression of miRNA-29b inhibits cell proliferation and decreases *DNMT3A* and *DNMT3B* mRNA, following reductions in the promoter methylation of ADAM metallopeptidase domain 23 (*ADAM23*) [103], cyclin A1 (*CCNA1*) [104], cyclin D2 (*CCND2*) [105], *CDH1* [106], cyclin-dependent kinase inhibitor 1C (*CDKN1C*) [107], cyclin-dependent kinase inhibitor 2A (*CDKN2A*) [108], HIC ZBTB transcriptional repressor 1 (*HIC1*) [109], Ras association domain family member 1 (*RASSF1*) [110], slit guidance ligand 2 (*SLIT2*) [111], TNF receptor superfamily member 10d (*TNFRSF10D*) [112], and tumor protein p73 (*TP73*) [113] tumor-suppressor genes, which improves breast cancer therapy [70]. In TNBC, the overexpression of miR-29B-1-5p inhibits the expression of DNMTs, followed by inhibiting the promoter methylation modification of tumor suppressor genes (TSG) secretoglobin family 3A member 1 (*SCGB3A1*/*HIN1*), Ras association domain family member 1 (*RASSF1A*) and *CCND2*, and inhibiting cell growth [114].

### 5.3. Hypomethylation of miRNA in Breast Cancer

The hypomethylation of aberrant miRNAs is uncommon in breast cancer, but it still affects cancer development. Aberrant oncogenic miRNAs may activate the oncogenic pathways by inducing demethylases (TETs) or inhibiting the expression of methylation transferases DNMTs, ultimately leading to the upregulation of oncogenic miRNAs and promoting cancer development. In an ERα-positive breast cell line, miR-375 is highly expressed and hypomethylated. It targets Ras-related dexamethasone-induced 1 (*RASD1*), inhibits ERα activation and induces cell proliferation [115]. The overexpression of the methyl-CpG-binding domain protein 2 (*MBD2*) in MCF-10A leads to the demethylation of miR-496, while the depletion of *MBD2* in MCF-7 and MDA-MB-231 inhibits the expression of miR-496 [116]. MiR-21 is an oncogenic miRNA that is upregulated in breast cancer. DNA methylation regulates the expression of miR-21 by knocking out the DNA demethylases *TET3*, and thymine DNA glycosylase (*TDG*) reduces the expression of miR-21 [117]. In young women with breast cancer, the promoter of miR-124-2 is usually demethylated, and therefore miR-124-2 is identified as a survival biomarker of breast cancer [118].

## 6. Aberrant Methylation of miRNAs Leads to Drug Resistance in Breast Cancer

The aberrant methylation of miRNAs is closely related to chemotherapy resistance, affecting cell cycle, DNA damage repair, apoptosis, stem cell transformation and mesenchymal transformation [119]. In drug-resistant breast cancer, aberrant tumor suppressor promoter hypermethylation and oncogenic miRNA hypomethylation leads to the dysregulation of oncogenic miRNA, ultimately leading to drug resistance.

### 6.1. Hypermethylation of miRNAs Leads to Drug Resistance in Breast Cancer

In breast cancer, the abnormal methylation of miRNA is one of the reasons for drug resistance, whereby the sensitivity of cells to drugs decreases, and this is mainly manifested as breast cancer proliferation, migration, invasion and colony formation, as shown in Table 5. In tamoxifen-resistant (TamR) MCF-7 cells, the methylation level of the miR-27b promoter region is significantly higher than that of tamoxifen-sensitive (TamS) MCF-7 cells. The re-expression of miR-27b restores drug sensitivity, inhibits invasion and reverses the EMT-like trait [120]. In addition, miR-29s, miR-132 and platinum-based chemotherapy affect DNA methylation [121]. Furthermore, endocrine resistance can be produced by the expression of miR-200b and miR-200c, while the anti-methylation drug 5-aza-dC+TSA can reduce the expression of miR-200 family members, thereby restoring the sensitivity of cancer cells to endocrine therapy [122]. MiR-320, as a mediator of chemoresistance, directly targets transient receptor potential cation channel subfamily C member 5 (*TRPC5*) and nuclear factor of activated T cells 3 (*NFATC3*), and the expression of miR-320a is regulated by the methylation of the promoter and ETS proto-oncogene 1 transcription factor (*ETS-1*) [123].

### 6.2. Hypomethylation of miRNAs Leads to Drug Resistance in Breast Cancer

In drug-resistant breast cancer, the aberrant hypomethylation of oncogenic miRNAs and the higher expression of miRNAs trigger oncogenic pathways, leading to the loss of breast cancer sensitivity to chemotherapeutic agents and promoting tumorigenesis and progression, as shown in Table 6. For example, tamoxifen reverses the EMT in TNBC cells via miR-200c demethylation [124]. It has been reported that the demethylation of miR-663 is related to the chemical resistance of breast cancer, including chemotherapy with doxorubicin, docetaxel and cyclophosphamide, and demethylation of the miR-663 promoter region upregulates the miR-663 expression level in breast cancer cells [125]. In addition, studies have found that downregulating miR-93 increases apoptosis in BC cells. A higher proliferation rate and lower apoptosis rate are observed in drug-sensitive cells with miR-93 overexpression. The methylation level of miR-93 is significantly low in drug-resistant cells with seven specific CpG sites of methylation [126].

## 7. Conclusions and Perspective

In this review, we highlight the oncogenic mechanisms of miRNA and DNA methylation in breast cancer. Oncogenic miRNA is highly expressed by inhibiting DNMTs or activating TETs, while suppressor miRNAs target DNMTs in breast cancer. Both the methylation of the miRNA promoter region and miRNAs’ regulating effects on DNMTs play important roles in breast cancer development and drug resistance, which may provide new therapeutic targets for drug resistance in breast cancer.

The two oncogenic mechanisms of miRNA and DNA methylation provide new strategies for breast cancer treatment. From February 2016 to April 2019, the General Surgery Department completed a clinical study with 20 unifocal and 20 multifocal breast cancer patients’ tissue samples, which were selected to analyze the expression of 84 microRNAs (Clinical Trials: NCT04516330). In December 2021, the Irish Cancer Trials completed a clinical trial to identify miRNA biomarkers that could be used to monitor patient response to chemotherapy and hormone therapy (Clinical Trials: NCT01612871). These clinical studies all suggest that specific miRNAs may be present in breast cancer patients with different types, stages and treatment cycles, and they may serve as biomarkers for individualized treatment. Aberrant miRNAs in breast cancer, including those caused by methylation, also appear to serve as biomarkers for breast cancer, offering possibilities for individualized breast cancer treatment.

With more understanding of the mechanisms of aberrant DNA hypermethylation in breast cancer, new therapeutic targets and new epigenetic treatment strategies will emerge. These targets for treatment may include DNMTs or other components of the DNA methylation machinery. Currently, there are two DNA Methyltransferase Inhibitors (DNMTis)—azacytidine and decitabine—approved for treating patients affected by myelodysplastic syndromes (MDSs) and acute myeloid leukemia (AML), while others are being trialed in multiple forms. Much effort is being made to improve efficacy through combining DNMTs with other therapies. Recent studies combining DNMTi with radiotherapy may provide a new treatment pathway for patients, especially for those who cannot tolerate platinum-based chemoradiotherapy [127].

Currently, there are many RNAi drugs. However, research into using miRNA-mimicking drugs in breast cancer clinical therapy is rare. Needless to say, the discovery of the importance of miRNAs in gene regulation and their association with cancer and other diseases has made them important targets for drug development. However, for nucleotide-based drugs, the greatest barriers to their action in vivo are degradation by nucleases and the escape of drug molecules from endosomes during endocytosis. The potential immunostimulatory effects and the lack of target specificity to the lesion area also represent other significant challenges for drug delivery systems [128]. The development of miRNA-targeted DNMT-induced tumors can be achieved by promoting the expression of miRNAs, such as small-molecule drugs, which may lead to miRNA therapies. However, this may trigger a series of unknown and unpreventable consequences due to the multitude of miRNA downstream target genes.

There is a degree of circular regulation of the oncogenic mechanisms between miRNA and DNA methylation. The promoter region of miRNA is methylated by DNMTs, while miRNA also targets DNMTs to inhibit its expression, such as *DNMT3B* inducing miR-200b promoter methylation in TNBC. On the other hand, miR-200b would inhibit the expression of *DNMT3B* involved in breast cancer invasion, migration and mammosphere formation [129]. However, little has been reported so far, and this cyclic regulation mode raises a great challenge in the use of DNA methylation inhibitors or small-molecule drugs for miRNA BC clinic treatment.

## Figures and Tables

**Figure 1 ijms-24-09360-f001:**
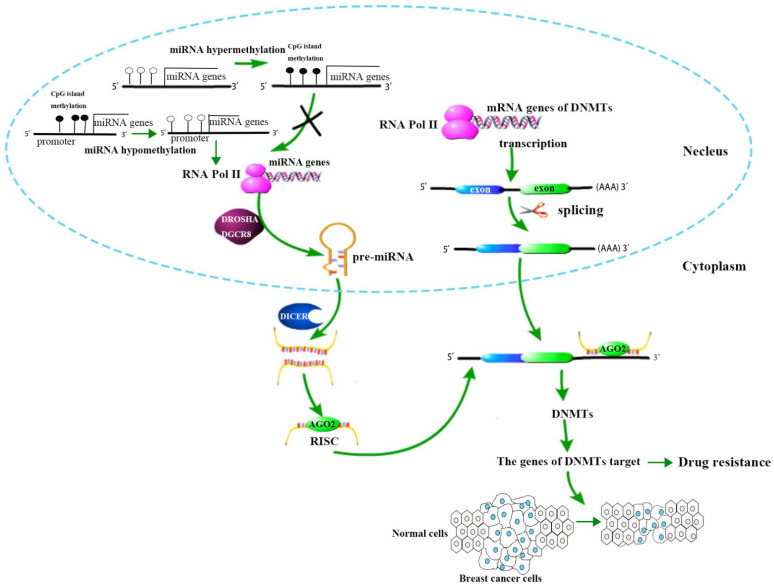
The regulatory mechanisms of miRNA and DNA methylation in breast cancer. A schematic involved in the biogenesis and function of miRNA. Genes of miRNAs are first transcribed by *pol II* as pri-miRNA. Then, this nascent pri-miRNA is cleaved and shortened by DROSHA and DGCR8 in the nucleus to produce pre-miRNA. The pre-miRNA is transports into the cytoplasm by Exp5, where it is processed, cleaved and modified by RNase III, Dicer/TRBP and AGO2 to generate miRNA duplexes. Finally, the double strand is dissociated into a single strand, and AGO-2 is connected with guide strand to form an RNA-induced silencing complex (miRNA:RISC). In addition, the DNMTs gene undergoes a series of complex processes of transcription and finally forms the mature mRNA. The methylation of miRNAs is usually regulated by DNMTs. In turn, miRNA:RISC pairs with its complimentary target sequence on the 3’-UTR of DNMTs mRNA in a perfect and imperfect manner to suppress the expression level of DNMTs, forming a regulatory loop between miRNA and DNMTs in BC development. Note: The arrows represent the steps involved in the biogenesis and functions of miRNAs.

**Table 1 ijms-24-09360-t001:** Aberrantly hypermethylated genes involved in drug resistance in BC and their function.

Genes	Drug Resistance	Function	Reference
*GFBI*	Trastuzumab	EMT	[54]
*BMP6*	Doxorubicin	EMT	[52]
*RASSF10*	Docetaxel	Cell proliferation	[57]
*SALL2*	Tamoxifen	Induction of ERα and downregulation of PTEN and activation of Akt/mTOR signaling pathway	[51]
*GCS*	Doxorubicin	Unknown	[49]
*PAX2*	Oftamoxifen	Unknown	[55]
*ESR1*	Endocrine, Cisplatin	Unknown	[50,56]

**Table 2 ijms-24-09360-t002:** Aberrantly hypomethylated genes involved in drug resistance in BC and their function.

Genes	Drug Resistance	Functions	Reference
*SH3GL2*	Doxorubicin, Epirubicin, 5-flurouracile, Cyclophosphamide	Proliferation	[61]
*MDR1*	Multi-Drug	Tumor size and advanced tumor stage	[59]
*MMP1*	Tamoxifen	Apoptosis	[60]
*LDHB*	Tamoxifen	Unknown	[58]

**Table 3 ijms-24-09360-t003:** Epigenetic miRNAs and their functions in breast cancer.

Epigenetic miRNA in Breast Cancer	Function	Reference
miR-29c	Proliferation	[72]
miR-200a	Proliferation	[73]
miR-200c/141	Invasion	[74]
miR-203	Invasion	[75]
miR-195/497	Proliferation, invasion	[76,77]
miR-296-5p/-512-5p	Proliferation, cell apoptosis	[78]
miR-31	Migration, invasion	[79,80]
miR-145	Migration, invasion, angiogenesis	[81]
miR-133a-3p	Proliferation, migration, invasion and stemness	[82]
miR-200b	Migration, invasion, stemness	[83]
miR-124a-1/2/3	Clinical makers (tumor growth, lymph node metastasis)	[84]
miR-1258	Clinical makers (lymph nodes or distant organs metastasis)	[85,86]
miR-9-3/339	Clinical makers (lymph node metastasis, late (III–IV) clinical stages, tumor size)	[86]
miR124-1/-34B/-34C	Clinical maker (late (III–IV) clinical stages)	[86]
miR-127	Clinical makers (lymph node metastasis, late (III–IV) clinical stages)	[86]
miR-132/-137	Clinical features (lymph node metastasis, tumor defferentiation, malignancy)	[85]
miR-205	Unknown	[87]
miR-124a3/-148/-152/-9-1/633	Unknown	[89]

**Table 4 ijms-24-09360-t004:** The DNMTs targeted by miRNAs and their functions.

miRNAs	miRNA Target DNMTs	Function	Reference
miR-152	*DNMT1*	Migration	[93]
miR-148a	*DNMT1*	Unknown	[94]
miR-143	*DNMT3A*	Proliferation	[95]
miR-101	*DNMT3A*	Proliferation, migration	[96]
miR-194	*DNMT3A*	Cell cycle	[97]
miR-770-5p	*DNMT3A*	Migration, invasion (EMT)	[98]
miR-29c-5p	*DNMT3A*	Unknown	[99]
miR-29c	*DNMT3B*	Proliferation, migration, invasion	[100,101]
miR-221	*DNMT3B*	Stemness	[102]
miR-148b/-26b/-29c	*DNMT3B*	Unknown	[101]
miR-29b	*DNMT3A*, *DNMT3B*	Proliferation	[70]
miR-29B-1-5p	*DNMT1*, *DNMT3A*, *DNMT3B*	Proliferation	[114]

**Table 5 ijms-24-09360-t005:** Aberrantly hypermethylated miRNAs involved in drug resistance in BC and their functions.

miRNA	Drug Resistance	Function	Reference
miR-27b	Tamoxifen	Invasion (EMT)	[120]
miR-29s/132	Cisplatin	Proliferation, migration, invasion	[121]
miR-200b/c	Tamoxifen and Fulvestrant	Proliferation, migration, invasion	[122]
miR-320a	Adriamycin and Paclitaxel	Unknown	[123]

**Table 6 ijms-24-09360-t006:** Aberrantly hypomethylated miRNAs involved in drug resistance in BC and their functions.

miRNA	Drug Resistance	Function	Reference
miR-200c	Tamoxifen	Invasion (EMT)	[124]
miR-663	Doxorubicin, Docetaxel and Cyclophosphamide	Proliferation, apoptosis	[125]
miR-93	Doxorubicin	Proliferation, apoptosis	[126]

## Data Availability

Not applicable.

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
