# Peer review of "The Mechanism of DNA Methylation and miRNA in Breast Cancer"

_ijms, 2023, doi:10.3390/ijms24119360_

Round 1

Reviewer 1 Report

Authors provide a comprehensive review of the role of microRNAs (miRNAs) and DNA methylation in breast cancer development, progression, and drug resistance. The review highlights the complex interplay between oncogenic miRNAs and DNA methylation machinery, including the regulatory effects of miRNAs on DNA methyltransferases (DNMTs) and ten-eleven translocation (TET) enzymes. The review discusses how oncogenic miRNAs can be upregulated by inhibiting DNMTs or activating TETs, while tumor-suppressor miRNAs target DNMTs in breast cancer.

The paper also explores the significance of aberrant methylation of miRNAs in breast cancer, focusing on the effects of hypermethylation and hypomethylation on cancer development and drug resistance. Authors delve into the potential of using aberrant methylation of miRNAs as biomarkers for cancer detection and prediction of clinical outcomes. Moreover, the paper emphasizes the importance of understanding the mechanisms of aberrant DNA hypermethylation in breast cancer to develop new therapeutic targets and treatment strategies.

The review briefly mentions the current status of DNA methyltransferase inhibitors (DNMTis) in clinical use and the challenges faced in developing miRNA-targeted therapies. Despite the potential of miRNA-based therapies, the paper acknowledges the difficulties in drug delivery and target specificity, as well as potential immunostimulatory effects. The circular regulation between miRNA and DNA methylation is discussed, highlighting the challenges in using DNA methylation inhibitors or small molecule drugs for miRNA-based breast cancer treatments.

Overall, this paper offers an in-depth analysis of the complex relationship between miRNAs and DNA methylation in breast cancer, emphasizing the potential of these epigenetic mechanisms as therapeutic targets and diagnostic biomarkers. The review provides a solid foundation for further research into the development of miRNA and DNA methylation-based therapies for breast cancer. I would recommend this article for publication.

Nevertheless, I have a few minor comments:

1) While the focus of the paper is on miRNA and DNA methylation, it would be helpful to briefly mention other epigenetic mechanisms that might be involved in breast cancer development and progression, such as histone modification and chromatin remodeling.

2) While the paper mentions some potential therapeutic targets and strategies, it would be helpful to discuss ongoing or completed clinical trials that have investigated the use of miRNA or DNA methylation-based therapies in breast cancer. This information would help to contextualize the potential impact of these approaches in clinical settings.

3) Since breast cancer is a heterogeneous disease, discussing the potential of using miRNA and DNA methylation signatures for patient stratification and personalized medicine could provide insights into the clinical utility of these epigenetic markers.

4) Section 3.2 and 3.1 will benefit from a table (gene - methylation status - treatment affected)

5) Line 41: The sentence seems to contain a structural error: "The therapy combination as a result of high heterogeneity, high rates of metastasis, poor prognosis, and lack of therapeutic targets for TNBC." Suggestion: "The therapy combination faces challenges due to the high heterogeneity, high rates of metastasis, poor prognosis, and lack of therapeutic targets for TNBC."

6) Line 62: "were" - are

7) Line 314: "out of methylation" - demethylated

8) Line 161: "expressed in patients with advanced breast cancer patients"

9) Line 194: "epigenetic" - epigenetically

There are quite a few mistakes that have to be addressed: typos, wrong tenses, structure errors.

Author Response

详情请参阅附件。

Reviewer 2 Report

Comments on the manuscript by Ma et al.,

In this manuscript, Ma et al. review the role of miRNAs and methylation in the pathogenesis of breast cancer. The manuscript is pertinent and attractive to the field, but multiple corrections and clarifications are required, listed below.

-Lines 29, 30. Importantly, breast cancer is the most frequently diagnosed tumor in humans (if basal cell carcinoma and skin squamous carcinoma are excluded) (this year, it surpassed lung cancer) and is also the leading cause of death in women (not the fifth). Please check and seek proper references.

-Lines 30-33. Intrinsic breast cancer subtypes are based on gene expression originally done by expression arrays. Please look for the correct references.

- Line 33. I do not think the "basic-like" tumor exists. Please revise

- Lines 33-35. To our knowledge, treating luminal B tumors is no alternative to receiving only endocrine therapy. He always needs chemotherapy. Please review.

-Lines 41-42. The sentence seems incomplete.

-Lines 52, 54. In this introduction, I believe it is not pertinent to randomly name Alzheimer's and congenital heart disease between only two tumors. Instead, it would be better to indicate other tumors where methylation is relevant in their molecular pathogenesis. Only breast and colon appear. You can list several subtypes and end up with tumors in general;  practically, it can be said that aberrant methylation participates in the pathogenesis of all tumor types.

-Genes should be italicized throughout the text—also, the words with origin in Latin as in situ.

-Line 58. Question: I was wondering whether there can be no specific hypomethylation of genes. Is it just only for the whole genome?

-Lines 61, 62. I do not understand that the methylation sites are located upstream and downstream of CpG dinucleotides; it is the C of the CpG itself that is methylated (5- methylcytosine).

-Lines 68, 69. I think I do not understand what you mean here. In what sense does methylation prevent the diagnosis of BC?

-Line 71, 72. What do you mean by "invasive tissue components"?

-Lines 107, 108. A very general sentence remains when saying "the promoter"; perhaps we should say "the promoters of certain genes."

-Lines 108, 109. What is the mechanism?

-Lines 119-121. We are not very clear about what it means. Please, clarify what it has to do with methylation.

-Line 172. I would put the title of Figure-1 in bold.

-Line 172. Better plural: "MiRNAs are ..."

-Figure-1: Please, write DROSHA and DGCR8 in white or outside the illustration because the letter in black makes it difficult to read.

-Line 180. I think "are" would be missing before "usually"?

-Line 182. I think that "from" should be changed to "form."

-Lines 195, 196. I guess it means they can be clinical markers (not that "are involved... and clinical markers"). Please rewrite.

-Lines 201-204.  The inhibition of MET and MET TK and, above all, the inhibition of EGFR is doubtful that they lead to proliferation after their inhibition. Please revise.

-Lines 219 and 221. When it says "BC tumors," it is a redundancy: C (cancer) and tumors. If you say BC, you would not need to add tumors.

-Line 234, where it says "according literatures" we think it should say "according to the literature."

-Lines 233, 234. I do not understand what is meant by those miRNAs having only one function.

-Lines 245-251. Although these miRNAs' hypermethylation function in BC is unknown, perhaps something can be said about their function in other tumor contexts.

-Line 253. In the context of tumors, it may be better to say "... to deregulate DNA methylation."

-Lines 266-268. It is recommended to include references in each of these molecules in which it is indicated that they are suppressors.

-Line 270. CDH1 is "an E-cadherin encoding gene." As written, it seems that there are more genes encoding E-cadherin. Perhaps I am wrong, but I think that would not be the case.

-Lines 289-294. Please, include references that show that they are suppressors. In some cases, it would be more or less controversial.

-Lines 303, 304. I would write "oncogenic pathways" in plural because there is not one but many of them.

-Lines 306-308. According to ref 90 in its abstract, it says that ER-alpha is reduced and favors cell growth. Please, revise that paper and the sentences written in the manuscript.

-Line 320. Where it says "with oncogenic miRNA hypermethylation..." We think you meant "hypomethylation"; is it correct?

-I would doubt the exhaustiveness of the compilation in this section. Are only four hypermethylated miRNAs associated with drug resistance in BC (Table 3)? The same is true for Table-4. Although not all were commented on the text, it would be good to check and expand the tables if possible.

-Lines 338, 338. We think "Function" should be plural.

-Line 340. Start the text with a capital letter.

-Lines 341, 342. Check to avoid repetitions of words (oncogenic miRNAs).

-Line 369. Put "inhibitors" in the plural.

I think the manuscript could benefit from English editing.

Author Response

Please refer to the attachment for more details.
